# Intranasal Peptide Therapeutics: A Promising Avenue for Overcoming the Challenges of Traditional CNS Drug Development

**DOI:** 10.3390/cells11223629

**Published:** 2022-11-16

**Authors:** Meenakshi Bose, Gabriela Farias Quipildor, Michelle E. Ehrlich, Stephen R. Salton

**Affiliations:** 1Nash Family Department of Neuroscience, Icahn School of Medicine at Mount Sinai, New York, NY 10029, USA; 2Department of Neurology, Icahn School of Medicine at Mount Sinai, New York, NY 10029, USA

**Keywords:** intranasal, peptide therapeutics, IGF-1, BDNF, oxytocin, PACAP, insulin, NAP, GLP-1, NBD, NPY, GDNF, EPO

## Abstract

The central nervous system (CNS) has, among all organ systems in the human body, the highest failure rate of traditional small-molecule drug development, ranging from 80–100% depending on the area of disease research. This has led to widespread abandonment by the pharmaceutical industry of research and development for CNS disorders, despite increased diagnoses of neurodegenerative disorders and the continued lack of adequate treatment options for brain injuries, stroke, neurodevelopmental disorders, and neuropsychiatric illness. However, new approaches, concurrent with the development of sophisticated bioinformatic and genomic tools, are being used to explore peptide-based therapeutics to manipulate endogenous pathways and targets, including “undruggable” intracellular protein-protein interactions (PPIs). The development of peptide-based therapeutics was previously rejected due to systemic off-target effects and poor bioavailability arising from traditional oral and systemic delivery methods. However, targeted nose-to-brain, or intranasal (IN), approaches have begun to emerge that allow CNS-specific delivery of therapeutics via the trigeminal and olfactory nerve pathways, laying the foundation for improved alternatives to systemic drug delivery. Here we review a dozen promising IN peptide therapeutics in preclinical and clinical development for neurodegenerative (Alzheimer’s, Parkinson’s), neuropsychiatric (depression, PTSD, schizophrenia), and neurodevelopmental disorders (autism), with insulin, NAP (davunetide), IGF-1, PACAP, NPY, oxytocin, and GLP-1 agonists prominent among them.

## 1. Introduction

Central nervous system (CNS) disorders, from developmental to degenerative, provide challenges to the generation of effective therapeutics. Despite the increasing incidence and improving diagnosis of diseases (such as depression, Alzheimer’s, and autism), there has been significant pruning of CNS drug development ventures by the pharmaceutical industry due to the high failure rates of candidates in clinical trials (approaching 100%) [1,2,3]. CNS drugs have high failure rates and significantly longer times in development, which is attributed to complications in drug transport across the blood-brain barrier (BBB) [4,5]. The BBB maintains essential brain homeostasis through tight and adherens junctions formed by cell adhesion molecules between capillary endothelial cells, presenting a unique challenge to the effective treatment of many CNS disorders.

The current neuro-pharmaceutical market is dominated by lipid-soluble small-molecule therapeutics, primarily due to their ability to cross the BBB to varying degrees via passive/transcellular or carrier-mediated diffusion. However, increasing lipophilicity to improve brain penetration risks a dramatic reduction in solubility once in the aqueous environment of the brain’s interstitial fluid, and CNS-to-blood transporters further prevent drug accumulation within the brain [6]. Additionally, traditional oral and systemic administration strategies have significant off-target effects in the periphery, and peptide/protein therapeutics delivered via this route are vulnerable to proteolytic degradation and rapid clearance by the liver and kidneys, limiting transport across the BBB and CNS membranes [7].

Over the last several decades, following the sequencing of the human genome and with increased dependence on proteomics- and bioinformatics-based approaches, there has been increasing interest in targeting protein-protein interactions (PPIs) [8]. PPIs regulate many essential cellular processes, forming the complex networks known as the “interactome”. Aberrant PPIs are implicated in the development of disease states, particularly aggregation-associated diseases, such as Alzheimer’s disease (AD) and Parkinson’s disease (PD) [9]. The lack of effective small-molecule drugs to disrupt interactions between proteins has led to PPIs being viewed as challenging or “undruggable” targets [10]. Small molecule drug discovery approaches mainly focus on protein-ligand interactions that have a much smaller contact area in comparison to the larger “featureless” interface areas commonly found with PPIs, which also lack the traditional endogenous small molecular ligands used for reference in small-molecule drug discovery. Drugs that have been found to act on PPIs have a higher molecular weight (>400 Da) than traditional small molecule drugs, leading to renewed interest in the development of therapeutic peptides, which have an intermediate weight between larger biologics and small molecules and can often very closely or exactly mimic the effects of the natural ligand [9,11].

In order to overcome the poor oral bioavailability of peptides, there have been many modern developments in non-oral drug delivery methods, including intranasal (IN) delivery (nose-to-brain) [5,12,13]. IN-administered compounds bypass the BBB through anatomical gaps in the skull in the nasal olfactory epithelium that allow neural connections to the olfactory bulb. Here, peptides access the brain through olfactory epithelial cells and then further along the olfactory and trigeminal nerve pathways for more distal brain delivery in a non-invasive manner (Figure 1) [14]. The exact mechanisms, sites, and pathways of action of CNS-targeted IN therapeutics are still under debate, but IN drugs and tracer molecules appear to be distributed throughout the brain via the CSF and extracellular fluid of the perivascular spaces around the olfactory and trigeminal nerves [14]. For excellent discussions of new advances in peptide and biomolecule formulation, leading to increased efficiency of IN olfactory adsorption, uptake, and stability, the reader is referred to the following reviews and articles [15,16,17,18,19,20,21]. Lastly, IN-administered peptides have the potential to directly enter the systemic circulation, avoiding first pass metabolism in the liver, so in certain limited cases peripheral plasma peptide pharmacokinetics have been studied and are well-reviewed here [16,17,22], providing additional insight into the potential for off-target effects [16] and also emphasizing the importance of the local intranasal mechanisms of peptide degradation [23].

IN administration as a method of bypassing the BBB and overcoming poor oral bioavailability has facilitated the pharmacotherapeutic development of peptides and other larger biologics for CNS disorders over the last two decades. Here, we comprehensively review the literature on IN administration of 12 different peptides in various stages of development for the treatment of neurodevelopmental, neuropsychiatric, and neurodegenerative diseases, the receptor systems or binding proteins that these peptides modulate, and the available preclinical experimentation and clinical trial data that have been used to measure their therapeutic efficacy to date.

## 2. Brain-Derived Neurotrophic Factor

Brain-derived neurotrophic factor (BDNF; MW: 14 kDa) is a growth factor in the neurotrophin family that includes nerve growth factor (NGF) and several other neurotrophins (NT-3, NT4/5, and NT-6). BDNF binds with high affinity to the tropomyosin receptor kinase B (TrkB), which activates multiple downstream signaling pathways, including Ras/mitogen-activated protein kinase (MAPK)/extracellular-signal-regulated kinase (ERK) and insulin receptor substrate 1 (IRS-1)/phosphatidylinositol-3-kinase (PI3K)/AKT. BDNF plays an essential role in neuronal survival, growth, and plasticity, as well as in learning and memory. Outside of the CNS, BDNF has also been shown to have roles in the angiogenesis and survival of adult endothelial cells, vascular smooth muscle cells, and cardiomyocytes. Due to BDNF’s highly regulated expression, great variability in BDNF levels has been observed in healthy individuals, as well as in patients suffering from a wide variety of disorders. Pathological dysregulation of BDNF is common to many psychiatric and neurodegenerative disorders, making it an indeterminate biomarker of specific disease but an increasingly valid biomarker of disease progression.

### 2.1. IN-BDNF Therapies for Neuropsychiatric Disorders

#### Major Depressive Disorder

BDNF’s dysregulation and depletion, particularly in the hippocampus, is well-demonstrated in both postmortem human brain studies of major depressive disorder (MDD) patients and animal models of depression. As such, development of BDNF as an MDD therapeutic has been of much interest, but though a diverse array of approaches has been used to administer BDNF for brain targeting (microinjection, intracerebroventricular injection, minipumps, etc.), there has been little therapeutic success.

Preclinical: With increased knowledge of the nose-brain pathway and IN therapeutics, IN-BDNF to mouse models of depression is being explored: in a chronic mild stress (CMS) induced-depression model, mice were given AAV-packaged BDNF fused with cell-penetrating peptides, human immunodeficiency virus-1 transactivator of transcription (TAT), and influenza A hemagglutinin glycoprotein 2 (HA2). CMS-exposed mice were given IN BDNF-HA2TAT/AAV for periods of 1, 5, and 10 days, followed by a 10 day wait period, before behavioral assessment and biochemical analysis to determine anti-depressant efficacy. Indeed, 10-day administration was found to be the most efficacious, though effects were seen after 5 days on tail suspension and in the forced swim test, and biochemical analysis showed increased hippocampal BDNF levels [26]. Additional studies were done in a post-stroke rat model of depression, as depression is the most common neuropsychiatric consequence of stroke, and studies have shown that decreased serum BDNF in post-stroke patients can be an early indicator of depression. A right middle cerebral artery occlusion was performed to establish stroke, following which placebo or BDNF-HA2TAT/AAV (20 µL/day) were administered in the final 10 days of a chronic unpredictable mild stress paradigm, followed by behavioral and biochemical analyses. Body weight loss was greatest in placebo and control (stress paradigm only, and stroke only) groups, and BDNF-HA2TAT/AAV treated groups had significantly higher body weights, as measured over a period of 4 weeks. Placebo and control groups demonstrated significantly increased immobility time in the forced swim test, which was reversed with IN-BDNF. BDNF expression was also found to be significantly lower in the prefrontal cortex and hippocampus of the stroke + stress and stress only rats, which increased with BDNF administration [27]. Collectively, these results suggest the promising efficacy of IN-BDNF as a therapeutic in multiple rodent models (and contexts) of depression.

## 3. EPO

Erythropoietin (EPO; MW: 30.4 kDa) is a glycopeptide hormone produced by the kidney to stimulate the proliferation and differentiation of red blood cells (RBCs) from erythrocytic progenitors, as well to stimulate other body tissues to protect and maintain crucial RBCs. When blood oxygen concentrations are within the normal range, EPO is synthesized predominantly by fibroblasts in the inner cortex of the kidney. However, when there are significant drops in blood O_2_ (i.e., hypoxia), interstitial cells in almost all zones of the kidney produce EPO [28]. EPO and its receptor (EPO-R) are also expressed in the brain in a hypoxia-sensitive manner to stimulate neural progenitor cell (NPC) proliferation and differentiation, as well as neuronal survival in a neuroprotective ischemic preconditioning mechanism. Brain-derived EPO has been shown to act locally via paracrine or autocrine signaling as a neuroprotective factor, additionally stimulating neural progenitor cells, neurons, glial cells, and endothelial cells in an analogous manner to its actions on erythrocytic progenitors outside of the nervous system [29].

### 3.1. IN-EPO Therapies for Neurodegenerative Disorders

#### Alzheimer’s Disease

AD is not only characterized by neuronal and glial dysfunction but also vascular dysfunction. Postmortem human AD and mild cognitive impairment (MCI) brain tissue show decreased levels of EPO. Therefore, harnessing the neuroprotective and neurogenic effects of EPO has been of much interest and its therapeutic delivery is currently being explored utilizing multiple routes of administration, including IN.

Preclinical: In a comparative study assessing the protective benefits of IN-EPO versus invasive intracerebroventricular (ICV) administration in an Aβ_25–35_ non-transgenic mouse model of AD, ICV-EPO at a dose of 125–500 µg/kg, IN-EPO at a much lower dose of 62–250 µg/kg, and low sialic form termed ‘Neuro-EPO’ (also intranasal) significantly prevented Aβ-induced learning deficits, hippocampal neurodegeneration, and reduced levels of neuroinflammatory markers tumor necrosis factor (TNF) and interleukin-1β (IL-1β) [30]. Subsequent studies in a transgenic amyloid precursor protein (APP) mouse model of AD (APP_Swe_) administered 125 µg/kg IN-Neuro-EPO to 12 and 14 month old mice 3 times per week for a period of 3 weeks, finding that IN-Neuro-EPO alleviated deficits in motor responses and memory/learning impairments, and reduced microglial activation, hippocampal amyloid β (Aβ) deposition, and levels of TNF, Bax/Bcl2-ratio, and Fas [31].

## 4. GDNF

Glial-derived neurotropic factor (GDNF; MW: 30.4 kDa) is a trophic factor that regulates embryonic nervous system development, with crucial roles in midbrain dopamine neuron differentiation and significant effects on other neuronal subpopulations as well as on embryonic kidney development [32]. GDNF and other members of the GDNF-family ligands (GFLs) are produced and secreted by a wide variety of tissues, usually as the precursor preproGFL [33]. PreproGFL can be differentially posttranslational processed into GDNF, neurturin, artemin, and persephin [33]. GNDF exerts its effects by binding to the GDNF family receptor tyrosine kinase (GFRα-1). GDNF is necessary for the maintenance of neuronal morphology as well as homeostatic signaling patterns, and has been shown to protect dopaminergic neurons from toxic damage [34].

### 4.1. IN-GDNF Therapies for Neurodegenerative Disorders

#### Parkinson’s Disease

GDNF has been a focus in PD research because it plays significant roles in the growth, regeneration, and survival of the substantia nigra (SN) dopamine. However, its inability to cross the BBB has been a significant obstacle to achieving efficacy, with clinical trials utilizing direct intracerebral infusions of GDNF to the striatum concluding with limited efficacy, in part due to the challenges with invasive brain infusions [35].

Preclinical: In a preliminary preclinical study in a 6-hydroxydopamine (6-OHDA) rat lesion model, rodents were pretreated with 10, 50, or 150 µg IN-GDNF prior to unilateral 6-OHDA injections. Both 50 and 150 µg IN-GDNF prevented 6-OHDA induced weight loss, reduced lesion severity by >40%, and prevented dopaminergic cell loss. In a subsequent group given three 50 µg IN doses (1 day before lesioning, 1 h before lesioning, 1 day post-lesioning), lesioned rats had even greater neuroprotective and regenerative effects, showing that multiple smaller doses in short intervals were more effective than a larger single dose [36]. In subsequent studies, IN-GDNF was administered in chitosan-coated nano structured lipid carriers modified with cell-penetrating TAT peptides (NLC-TAT-GDNF), as well as unmodified GDNF and placebo to MPTP mouse models of PD. Three weeks of treatment with 2.5 µg NLC-TAT-GDNF reversed motor deficits in rotarod testing, recovered dopaminergic neurons to healthy levels, and attenuated microglial hyperinflammation [37,38].

## 5. GLP-1

Glucagon-like peptide 1 (GLP-1; MW: 3.29 kDa) is a 30-amino acid peptide most commonly known for its role in lowering of postprandial glucose levels and in appetite regulation, but also with emerging roles in CNS stress responses and sympathetic nervous system regulation [39]. GLP-1 is first produced as proglucagon, primarily by intestinal epithelial endocrine L-cells in the periphery, as well as by preproglucagon (PPG) neurons in the CNS [39]. GLP-1 mediates its effects through binding at a transmembrane receptor (GLP-1R), found on pancreatic β cells and CNS neurons. Central GLP-1 secretion by PPG neurons can increase as part of a wider response by the CNS to emotional stress or conditioned affective food responses (visceral malaise), activating the sympathetic nervous system and HPA axis [39].

### 5.1. IN-GLP-1 Therapies for Neurodegenerative Disorders

#### Alzheimer’s Disease

As AD shares many pathological characteristics with type 2 diabetes mellitus and is also characterized with impairments in circadian rhythm, harnessing of GLP-1 for its neuroprotective effects has been a recent focus, particularly with the repurposing of a type II diabetes drug and GLP-1 analogue, exendin-4.

Preclinical: A preclinical study looking at the protective benefits of exendin-4 against Aβ_31–35_-induced disruptions of learning, memory, and circadian rhythm found that a single dose of IN-exendin ameliorated circadian rhythm-associated disturbances in locomotor activity and memory impairment [40]. A subsequent study examined IN-exendin-4 benefits on severe MCI in a senescence-accelerated mouse (SAMP8) model. Male mice were administered a combination of 8 international units (IU) IN-insulin, 5 µg IN-exendin, in a solution containing 2 mM of L-penetratin, a cell-penetrating peptide designed to boost the uptake of both therapeutic agents, or placebo, daily for 30 days. Improvements in spatial learning ability and memory and reduced Aβ deposition were observed in mice given IN-exendin/insulin [41].

## 6. Insulin/Insulin-like Growth Factor 1

Insulin (MW: 5.8 kDa) and insulin-like growth factor-1 (IGF1; also known as somatostatin C; MW: 7.65 kDa) are peptide hormones sharing significant structural homology with significant contributions to the regulation of whole-body metabolism and promotion of growth/cell renewal [42]. Within the CNS, insulin and IGF-1 are both implicated in neuronal survival/protection, synaptic maintenance and synaptogenesis, dendritic arbor development, and neuronal circuitry formation/regulation for CNS responses to environmental and peripheral tissue stimuli [43].

Early and/or increased risks of insulin resistance underlie several major neurodevelopmental disorders (NDD), with decreased CNS responsiveness to insulin contributing to developmental delays and cognitive impairments. In schizophrenia (SZ), Down Syndrome, and bipolar disorder (BD), increased brain insulin resistance and reduced IGF-1 signaling drive allostatic load/overload, leading to physiological dysregulation and the presentation of more severe symptoms throughout the body [44,45,46].

### 6.1. IN-Insulin or IN-IGF1 Therapies for Neurodevelopmental Disorders

#### 6.1.1. Autism Spectrum Disorders

Phelan-McDermid Syndrome [PMS; under the Autism Spectrum Disorders (ASD) umbrella] is a rare genetic condition caused by deletion or de novo pathological variation of the long arm terminal end of chromosome 22 (location q13.3; also referred to as 22q13.3 deletion syndrome) within the *Shank3* gene [47]. *Shank3* encodes for multidomain scaffolding Shank family proteins critical for postsynaptic function of glutamatergic synapses. *Shank3* deficiencies also account for 1–2% of all ASD cases, making it one of the most prevalent ASD mutations. This loss results in deficits in synaptic function, plasticity, and clinically presents with neural malformations, hypotonia, developmental delays, delayed/absent speech, autistic-like behavior, and dysmorphic facial features [48].

Clinical: Initial pilot clinical investigations into insulin and IGF-1 for the treatment of PMS utilized intraperitoneal injections of IGF-1 to treat nine children (aged 5–15) over a 3-month treatment period, with a double-blind, placebo-controlled, crossover design. Compared to placebo, children treated with IGF-1 showed significant improvement in social impairment and restrictive behaviors [49]. This early success led to investigations of IN-insulin on the development and behavior of PMS patients. In a randomized, double-blind, placebo-controlled clinical trial of 25 children (1–16 years) treated with daily IN-insulin sprays for a period of 6 months, IN-insulin improved developmental functioning by 0.4–1.4 months per 6 month period, with significant cognitive and social improvement in children <3 years of age, where PMS patients usually show delays in developmental growth [50].

#### 6.1.2. Bipolar Disorder

Insulin resistance is present in 52% of BD patients and is associated with a chronic course, non-responsiveness to treatment, adverse brain changes, further cognitive impairment, and an increased mortality rate [51].

Clinical: It was hypothesized that in stable euthymic individuals with bipolar disorder I/II (BD I/II), IN-insulin would enhance hippocampal-dependent neurocognitive function. Initial double-blind, placebo-controlled crossover trials evaluating the effect of IN-insulin on adult individuals with BD I/II (*n* = 62) utilized an adjunctive insulin treatment of 40 IU (*n* = 34) or placebo (*n* = 28) four times daily for a period of eight weeks. Neurocognitive function and outcome were assessed with neurocognitive battery testing; significant improvements versus placebo in executive function were seen with several tests. The authors noted a need to phenotype and or/genotype subjects based upon other possible pre-existing risk factors [e.g., apolipoprotein E (ApoE)] for further possible testing [52].

#### 6.1.3. Schizophrenia

SZ patients have been shown to have cognitive deficits in domains of attention, verbal memory, and executive function, with severity of impairment correlated to the extent of impact on real-life functioning. Additionally, alterations in brain insulin signaling pathways (particularly of PI3K/AKT) have been shown in SZ patients.

Clinical: Initial single-dose, double-bind, placebo-controlled trials were undertaken to evaluate the effect on verbal memory and attention deficits of IN-insulin in nondiabetic adult individuals (*n* = 30) with DSM-IV diagnoses of SZ or schizoaffective disorder and stabilized on another antipsychotic agent for at least one month. Participants performed cognitive tasks before and after receiving 40 IU IN-insulin or placebo. Despite decreased serum insulin levels, no significant differences in cognitive performance were observed [53]. Further repeated-dose randomized, double-blind, placebo-controlled studies (*n* = 45) were conducted over an 8-week period to assess the possible benefits of IN-insulin, with subjects receiving either 40 IU IN-insulin 4 times per day (*n* = 21) or placebo (*n* = 24). A battery of cognitive testing was used as a measure of psychopathology and cognition at weeks 0, 4, and 8. No beneficial effects were seen following IN treatment on psychopathology or cognition by any measures of testing [54]. Additional testing on the effects on body metabolism were done using whole body dual-energy X-ray absorptiometry and nuclear magnetic resonance (NMR) spectroscopy, for which 39 of 45 study participants were evaluated (insulin = 18, placebo = 21). Both measures failed to show a benefit of IN-insulin on any major metabolic outcomes, despite increased pro-opiomelanocortin (POMC) expression [55]. It has been concluded through these trials that although insulin signaling is impaired in the brains of SZ patients, centrally available insulin through IN delivery may not be able to overcome these impairments, leading to the observed lack of effect on cognitive, psychopathological, or whole-body metabolic measures.

### 6.2. IN-Insulin or IN-IGF1 Therapies for Neuropsychiatric Disorders

#### 6.2.1. Major Depressive Disorder

Clinical: Pilot trials to assess the therapeutic potential of IN-insulin on cognitive function and mood in adults with MDD have been done. Patients (*n* = 35) were given either 40 IU IN-insulin or placebo 4 times a day for a period of 4 weeks, after which they were switched to the other treatments (placebo or insulin, respectively) for another 4 weeks following a 1-week washout period. Neurocognitive tests were conducted at four time points: baseline phase 1 (week 0), endpoint phase 1 (week 4), baseline phase 2 (week 5) and endpoint phase 2 (week 9) [56]. No significant changes were seen in any of the cognitive or psychiatric measures used [56].

#### 6.2.2. Generalized/Social Anxiety Disorders

As IN-insulin reverses anxiety-like behavior in rodents, and rats deficient in brain insulin receptors exhibit anxiety and depressive-like behavior [57,58,59], IN-insulin has been explored in a few clinical trials for its ability to modulate anxiety and fear.

Clinical: In a clinical trial to assess IN-insulin’s ability to modulate HPA axis response to psychosocial stress, 26 male participants received a single dose of 40 IU IN-insulin prior to social stress testing. Plasma cortisol, saliva cortisol, heart rate, and blood pressure were measured at resting, baseline, and in response to testing. All measured parameters increased significantly in response to stress, and IN-insulin was able to significantly diminish saliva and plasma cortisol but did not affect heart rate or blood pressure under stress conditions. The authors concluded that a single IN-dose effectively lowers the stress-induced HPA axis response [60].

Recently, IN-insulin was administered in a double-blind, placebo-controlled differential fear-conditioning paradigm in 123 healthy participants. As social anxiety is the most common type of anxiety disorder, threatening social experiences were simulated with pictures of neutral faces followed by electric shocks. This paradigm was conducted in four phases over 3 days: acquisition (day 1), extinction (day 2), reinstatement and re-extinction (day 3). Participants were given either 160 IU IN-insulin or placebo on Day 2, 45 min prior to fear extinction. Participants given IN-insulin showed a significantly greater decrease of skin conductance response in comparison to placebo, though no other parameters changed significantly (fear startle, expectancy ratings) [61].

### 6.3. IN-Insulin or IN-IGF1 Therapies for Neurodegenerative Disorders

#### 6.3.1. Alzheimer’s Disease

Neurodegeneration in AD has not only been associated with an accumulation of Aβ and tau, but also with metabolic alterations in insulin signaling. AD patients have lower levels of insulin in the CSF and higher plasma insulin when compared to healthy controls, indicating peripheral insulin resistance in AD patients [62]. This brain insulin (and also IGF-1) resistance has been further demonstrated in the hippocampal formation and cerebellar cortex of AD patients, showing significantly decreased signaling in insulin receptor/IGF-1 receptor-IRS-PI3K pathways, in the absence of diabetes [63]. Impaired insulin signaling in the brain results in neuroinflammation, apoptosis, oxidative stress, and the overexpression of Aβ and tau, as reviewed by Nguyen et al. [64]. Therefore, restoring insulin levels in the CNS through non-invasive IN administration has been explored as a plausible therapeutic approach for AD cases, both in animal models and in human patients.

Preclinical: Several studies have investigated the effects of IN-insulin in WT and AD mouse models, showing dissenting results on cognition and signaling that were based on either acute or chronic insulin administration. For example, Gabbouj et al. showed that a single dose of IN-insulin in WT and APP/presenilin-1 (PS1) mice increased glucose uptake in the ventral brain and hippocampus of WT mice when compared to APP/PS1 mice, but did not improve spatial memory using the Morris Water Maze Test [65]. Nevertheless, using the same AD mouse model, another group demonstrated that 6 weeks of IN-insulin treatment improved cognitive deficits and brain insulin signaling while reducing Aβ production and plaque formation [66]. Similarly, long-term (2–6 weeks) IN-insulin administration in an AD rat model improved cognitive function and reduced tau hyperphosphorylation, inhibited microglial activation, and ameliorated neurogenesis deficits [67,68,69].

Clinical: The first randomized study of acute IN-insulin (20 IU and 40 IU) resulted in improved verbal memory in patients with AD or MCI but only in the absence of the apolipoprotein E (APOE) ε4 allele [70,71]. Given that ε4 negative AD patients had previously shown signs of insulin resistance [62], this study suggested that cognitive effects in response to intranasal insulin may indicate disrupted insulin metabolism in AD and MCI patients that do not carry the ε4 allele. Similarly, an acute, double-blinded, randomized, placebo-controlled, crossover study of a rapid-acting insulin analog (glulisine) has also shown diminished therapeutic response of IN-insulin in APOE ε4 carrier AD patients [72]. Another randomized placebo-controlled study showed that daily IN-insulin (20 IU) over 21 days improved cognition, measured by retention of new information after a delay, in patients with early stage AD [71]. However, their group reported improvement as “percentage change from placebo”, which could have affected the conclusions if there was a decrease in the placebo control group. Another group found no benefit on cognition when using Humulin-R as intranasal insulin (60 IU) administered four times daily over 48 h, although they did not stratify by APOE ε4 carriers; patients were supplemented with vitamin D2, and their dose may have been too high (IN-insulin response for memory is ∩-shaped), all of which could have potentially masked an effect [73]. Indeed, longer IN-insulin (20 IU) administration for a period of 4 months has shown improvements in delayed memory, and preservation of general cognitive abilities in patients with MCI or AD [74]. Patients in both insulin dose groups showed preserved glucose uptake in several areas of the brain, and the effects seemed to be stronger in AD compared to MCI patients [74]. In a subsequent study, this group also showed that there are important sex differences stratified by APOE ε4 carrier status. For instance, men that did not carry the APOE e4 allele showed cognitive improvement, while for women, cognitive outcome was the poorest when they were administered a high insulin dose (40 IU) [75]. The authors concluded that men were more sensitive to IN-insulin and have more pre-existing insulin abnormalities in the CNS than women [75], showing the importance of including both sexes in clinical trials.

Interestingly, IN administration of 40 IU long-acting insulin analog (detemir) improved peripheral insulin resistance during treatment, which was associated with improved verbal memory in AD patients that carry APOE ε4, unlike what had been previously reported with rapid acting insulin analogs [76]. Detemir also improved visuospatial and verbal working memory in all study participants that were administered 40 IU, but it increased peripheral insulin resistance in the APOE ε4 negative patients during treatment [76]. These differences were explained by the affinity of regular insulin to insulin receptors as well as the pharmacodynamics of the different formulations, given that long-acting insulin analogs result in greater cumulative exposure because of their increased half-life, while rapid-acting insulin mimics the post-prandial release, reaching a higher acute peak [76]. However, when comparing effects between regular insulin and detemir, it was found that detemir’s long-term efficacy decreased, while regular insulin continued to show improvements on memory in MCI and AD patients [77]. Although there were technical limitations that affected the end-point interpretations of the first phase 2/3 multisite randomized double-blind clinical trial, their results suggested an advantage in the AD Assessment Scale, improved Aβ42/40 ratios, and improved Aβ42/total tau protein in the insulin group [78]. They further assessed effects in this cohort and found that 12 months of IN-insulin reduced white matter hyperintensity volume progression, which is correlated with cognition in their studies [79].

#### 6.3.2. Parkinson’s Disease

Preclinical: IN-insulin has received recent attention in PD research due to the commonality of its dysregulation in AD, PD, and other ND or metabolic disorders, and has been explored preclinically in 6-OHDA lesioned rodent models of PD. Rats received 400 µg IN-insulin (~11.5 IU) daily for 2 weeks, beginning 24 h post-lesion, which significantly ameliorated motor impairments (improved locomotor activity) and protected dopaminergic neurons against 6-OHDA neurotoxicity without affecting body weight or blood glucose levels [80]. In another study, 6-OHDA PD rats were given a single lower dose of 3 IU IN-insulin as pretreatment either 4 days, 2 days, or 30 min before surgery. Following lesioning, rats were again given 3 IU IN-insulin 5 days/week for 4 weeks; even with this significantly lower dose from the previous study (~25%), IN-insulin still rescued 6-OHDA motor deficits in motor behavioral battery testing and protected against dopaminergic neuron loss [81]. A subsequent study, using an even lower dose (2 IU) treatment for 6 weeks, found that 6-OHDA affected not only dopaminergic neurons in SN, but also insulin signaling in the ipsilateral hippocampus through pathway disruption from SN, and IN-insulin was able to restore the function of this injured insulin signaling pathway. IN-insulin was again shown to improve motor behavior, and at this very low dose also protected SN dopaminergic neurons and restored levels of pAKT and pGSK3β in the ipsilateral hippocampus affected by 6-OHDA [82].

Clinical: Initial clinical trials for IN-insulin in treatment of PD and parkinsonian-type multiple system atrophy have begun in recent years. In a double blind, placebo-controlled pilot study, eight PD patients (*n* = 8, 2 women) and six healthy age-matched participants self-treated with 40 IU IN-insulin daily for 4 weeks; IN-insulin treated PD participants had a significant word count increase, decreased severity of parkinsonism and improved motor score. IN-insulin did not affect any of the other parameters measured for cognition, gait, or depression [83].

#### 6.3.3. Huntington’s Disease

Though conflicting results make it difficult to understand the exact role of the IGF-1/AKT pathway in Huntington’s Disease (HD), high plasma levels of IGF-1 have been correlated to cognitive decline in HD patients, and preclinical data has shown the protective benefits of IGF-1 in striatal neuronal culture and in R6/1 mice.

Preclinical: A pilot IN-IGF-1 study has been carried out using a YAC128 mouse model of PD, expressing human mutant Huntingtin (mHtt) with ~128 CAG repeats. Six-month old male YAC128 and wild-type mice were given 35 µg IGF-1 with an alternate-day IN dosing paradigm for two weeks. In motor testing, mice treated with IN-IGF-1 demonstrated reduced motor impairment and improved locomotor activity (rotarod and open field), significantly increased cortical and striatal PI3K/AKT/mammalian targets of rapamycin (mTOR) signaling, reduced phosphorylation of mHtt, reduced markers of CNS metabolic abnormalities, and increased cortical (but not plasma) levels of IGF-1 [84].

## 7. NAP

NAP (alternative names: A-L108, Davunetide, NAPVSIPQ, or CP201) is an eight amino-acid peptide (MW: 824 Da) derived from the active domain of activity-dependent neuroprotective protein (ADNP) [85]. Expression of ADNP is highest during maternal gestation, driving neural induction and differentiation by enhancing Wnt signaling [86]. NAP/ADNP exert neuroprotective effects through PI3K/AKT and MAPK/MEK1 pathways. NAP binds to microtubule end-binding proteins 1, 2, and 3 (EB1, EB2, EB3) and modifies the transcriptional signatures of the SWItch/Sucrose Non-Fermentable (SWI/SNF) chromatin remodeling complex [87]. Mutations/deficiencies in *Adnp* genes have been associated with disruptions of axonal transport through the destabilization of microtubules, leading to impaired mitochondrial and cellular function, as well as alternative—and pathological—splicing, hyperphosphorylation, and prion-like aggregation of the microtubule-associated tau protein. This can manifest with ADNP-associated ASD, SZ, and other forms of intellectual disability, and may contribute to the progression of several neurodegenerative disorders. [87,88]. NAP pharmacotherapy for neurodegenerative disease has been explored via multiple routes of administration, including intranasal [89], which has also been utilized for amyotrophic lateral sclerosis [90] and diabetes [91].

### 7.1. IN-NAP Therapies for Neurodevelopmental Disorders

#### 7.1.1. Autism Spectrum Disorders

Preclinical: IN-NAP administration to *Adnp*-haploinsufficient mice ameliorated olfactory and behavioral deficits, abnormal brain structures, and improved cognition [87]. IN-NAP treatment decreased expression of inflammatory cytokines such as TNF, IL-6, and IL-12, suggesting that ADNP may be involved in the regulation of immune activity, and alleviated irregular immune activation [92].

Additional studies of ADNP’s protein sequence revealed the SH3-binding domain for SHANK3 (previously discussed in Section 6.1.1 to account for 1–2% of all ASD patients and for Phelan-McDermid Syndrome) and also actin-binding domains, suggesting possible shared underlying mechanisms for disease dysfunction [93]. Truncation of these regions in mice impaired end-binding protein 3 (EB3) activity, resulting in reduced Tau-microtubule (MT) interactions and in tauopathy-like features similar to Adnp^+/−^ mice. The administration of IN-NAP to mice homozygous for the ASD-associate InsG3680 *Shank3* mutation normalized Shank3-Adnp-actin interactions, regulated *Shank3* mRNA transcripts in *Adnp*-deficient male mice, and reduced anxious/depressive and repetitive behaviors [93].

Most recently, novel ADNP syndrome mice have been developed using CRISPER-Cas9 gene editing technology to produce mice heterozygous for *Adnp* pTyr718* (Tyr), a paralog of the most common ADNP mutation, which have Tyr-specific sex differences. NAP or placebo was first administered to these heterozygous Tyr pups subcutaneously and then intranasally after 21 days of age until 2 months of age. NAP treatment ameliorated *Foxo3* deregulation, early-onset hippocampal and visual cortex tauopathy in males, prevented dendritic spine abnormalities, and corrected gut microbiome abnormalities. Behavioral and abnormal visual, auditory, and speech delays were corrected in heterozygous pups, gait impairments were reversed, and pups gained weight similar to wildtype pups [94].

#### 7.1.2. Schizophrenia

Preclinical: IN-NAP has been investigated in the context of multiple mouse models of SZ, specifically disrupted in schizophrenia 1 (DISC1) and stable tubule-only polypeptide (STOP) mutant mice. STOP proteins are part of a family of microtubule-associated proteins (MAPs). There have been linkages to allelic variation in STOP genes and altered STOP protein expression in SZ. STOP^−/−^ mice have significant synaptic and behavioral deficits and hypermotility. Heterozygotic STOP^+/−^ mice with similar behavioral and pathological profiles were used to evaluate IN-NAP. Mice received either 0.5 µg of IN-NAP or 10 mg/kg clozapine via intraperitoneal injection daily for 7–10 weeks. Open field behavioral testing showed significant path reduction with both clozapine (~60% reduction) and NAP (30% reduction). IN-NAP treatment completely reversed deficits in novel object recognition and discrimination tests, and the Morris water maze, in STOP^+/−^ mice [95,96].

DISC1 is a microtubule-regulated protein with mutated/disrupted function in SZ and other psychiatric illnesses [97]. DISC1 mutant mice have decreased complexity of dendritic arbors and decreased neurite outgrowth, pathologically similar to abnormalities seen in post-mortem brains of SZ patients [97,98]. IN-NAP was evaluated alongside doxycycline (blocks mutated gene expression) and risperidone (a frequently-used neuroleptic) as controls [99]. Both risperidone and NAP were able to improve cognitive function as measured by novel object recognition, with NAP improving cognitive function with twice the efficacy of risperidone. Additionally, NAP but not risperidone normalizes DISC1 mouse behavior in an elevated plus maze, indicating the significant reduction of anxiety; NAP and risperidone also normalize Foxp2 expression in the hippocampus, a gene with many SZ-associated SNPs.

Clinical: Initial repeated-dose, multicenter, double-blind, placebo-controlled trials have been conducted by Allon Therapeutics to assess the effects of NAP (AL-108) on cognition and functional capacity in patients (*n* = 63) with DSM-IV diagnoses of SZ or schizoaffective disorder who were stable for >1 month on first or second generation depot antipsychotics. Participants received doses of 5 mg NAP (*n* = 22), 30 mg NAP (*n* = 22), or placebo (*n* = 25) for a period of 12 weeks. A single significant change in functionally-meaningful cognition with NAP administration was observed. No further effects were observed, which the authors concluded might be due to an insufficiently large study for effect (50/group needed) [100].

### 7.2. IN-NAP Therapies for Neurodegenerative Disorders

#### 7.2.1. Alzheimer’s Disease

Preclinical: Some of the earliest work with IN-NAP, the active domain of ADNP, assessed its neuroprotective role in aged rodents, as well as in various AD mouse models. In rats treated with the neurotoxin cholinotoxin ethylcholine aziridium, which disrupts acetylcholine systems, animals that underwent a 5-day pretreatment with 5 µg daily IN-NAP had significantly better water maze performance for both healthy and cognitively-impaired rats, and NAP significantly protected against the loss of choline acetyl transferase activity [101]. In 8-month old healthy mice given 0.5 µg IN-NAP daily for a period of either 5 or 8 months, IN-NAP significantly reduced anxiety in the elevated plus maze with each dosing timeframe, and increased efficacy was noted with increasing treatment length [102]. In the APP/PS1 transgenic amyloidosis mouse model, 0.5 µg daily IN-NAP for 3 or 6 months reduced Aβ_40_ and Aβ_42_ isoforms and hyperphosphorylated and insoluble tau, and also improved cognitive performance in the Morris water maze [103,104]. In an ADNP-deficient mouse model (ADNP^+/−^), which presents with increased tau hyperphosphorylation, tauopathy-induced memory impairments, and neurodegeneration, 2 weeks of daily 0.5 µg IN-NAP to 2 and 9 month old mice partially ameliorated cognitive deficits and reduced tau hyperphosphorylation, showing that even short-term treatment can have significant effects on MCI/AD phenotypes at various stages of degeneration [105]. In a double-tauopathy mouse model (DM-Tau-tg; P301S, K257T), 0.5 µg IN-NAP daily for ~5 months improved Morris water maze performance to that of controls, increased levels of soluble tau, and decreased neurotoxic hyperphosphorylated tau [106,107].

Clinical: Initial phase-2 double blind, placebo-controlled, ascending dose clinical trials have assessed the safety, tolerability, and efficacy of NAP—under the name of ‘AL-108’ and managed by Allon Therapeutics—in MCI patients age 55–85, gender-inclusive. Patients either received 15 mg twice-daily IN-NAP or placebo for a period of 12 weeks, with limited adverse events noted (i.e., headache) and a cognitive battery carried out at baseline and after 4, 8, 12, and 16 weeks of IN treatment [108,109]. There were no significant improvements in overall composite cognitive memory scores, but a potential for protection was suggested based upon significant effects in both the delayed match-to-sample and digit span forward [109].

#### 7.2.2. Parkinson’s Disease

Preclinical: As NAP interacts with both neuronal and glial tubulin to modulate microtubule assembly and clears amyloid and hyperphosphorylated tau in AD and SZ mouse models, as noted above, IN-NAP was also investigated in a transgenic mouse model of PD overexpressing α-synuclein. Mice given 2 µg IN-NAP, 5 days a week for 24 weeks, had a significantly lower phospho-tau/tau ratio in the subcortical region, and mice given even higher doses of 15 µg/day also had a decreased phospho-tau/tau ratio in the cerebellum. Both dose groups had reduced hyperactivity, improved testing of habituation in a novel environment, but no improvements in severe motor deficits in motor battery testing [110]. In a subsequent study administering 2 µg IN-NAP daily for 2 months, NAP-treated mice did show a significant decrease in errors in the beam traversal test, and decreased α-synuclein aggregates in the SN with this alternate dosing strategy (7 days/week vs. 5 days/week) [111].

## 8. NBD

The 22 amino acid NBD peptide, corresponding to the NEMO-binding domain (NBD; MW: 199.5 Da) of the IKKα subunit of the canonical nuclear factor-κB (NF-κB)-activating IkB complex, inhibits canonical NFkB activation without inhibiting noncanonical (or basal) NF-kB activity [112,113]. Nuclear factor-κB (NF-κB) essential regulator (NEMO) is a regulatory subunit component of the multi-subunit IκB kinase (IKK) polyubiquitin complex, also containing IKKα and IKKβ catalytic subunits. NF-κB represents a family of inducible transcription factors that regulate many genes involved in multiple processes of innate and adaptive immune and inflammatory responses, making them a pivotal mediator of pro-inflammatory responses (e.g., cytokines, chemokines, inflammasomes) [114]. NEMO, IKKα, and IKKβ are synthesized in cells throughout the body in response to intracellular stress, inflammatory membrane receptor binding (e.g., TNF superfamily, toll-like, interleukin), or DNA/oxidative damage. NF-κB can be activated through either canonical or alternative signaling pathways, both of which are important in the regulation of inflammation [114,115]. The IKK complex, when activated, binds and degrades NF-κB inhibitor Iκβα by site-specific phosphorylation, thus activating the canonical signaling pathway for NF-κB. Deregulated NF-κB activation contributes to the pathogenic processes of many ‘hyperinflammatory’ diseases and disorders.

### 8.1. IN-NBD Therapies for Neurodegenerative Disorders

#### Alzheimer’s Disease

Chronic hyperactivation of the canonical pathway for NF-kB, a key mediator of glial function and neuroinflammation, is significantly associated with the development of AD-type dementia, and genetic suppression of this pathway in early onset neurodegeneration models rescues pathology [116]. Harnessing this pathway has therefore been of significant therapeutic interest.

Preclinical: In the triple-transgenic 5xFAD mouse model of AD (B6SJL background), with demonstrated NF-kB hippocampal hyperactivation, IN-NBD (0.1 mg/kg) was administered daily for 1 month in 5-month-old male mice. IN-NBD significantly inhibited hippocampal NF-kB activation and neuroinflammation, and reduced plaque formation and neurodegeneration. IN-NBD also significantly upregulated several neuroplasticity-related molecules (CREB, GluR1) and improved spatial learning and memory in the Barnes and T Mazes, as well as in novel object recognition [117].

## 9. NGF

Nerve Growth Factor (NGF; MW: 26.9 kDa) is a neurotrophin essential for the development and maintenance of neurons in the CNS, particularly forebrain cholinergic neurons, and in the peripheral nervous system. In the CNS, the highest expression levels of NGF are found in the cortex, hippocampus, and pituitary, with lower but still significant levels in the basal ganglia, thalamus, and spinal cord. NGF is a critical growth factor during development and in adulthood, regulating differentiation and survival of neural crest-derived sympathetic and sensory neurons. NGF binds to the tropomyosin kinase receptor A (TrkA) and to the p75 neurotrophin receptor (p75^NTR^), which activate MAPK, ERK, PI3K, PLC-γ, and regulate cell survival/cell death pathways. Through these actions, NGF controls attention, arousal, motivation, memory, autonomic responses, and the stress axis [118].

### 9.1. IN-NGF Therapies in Neuropsychiatric Disorders

#### Major Depressive Disorder

MDD patients have reduced serum levels of NGF and post-mortem analysis of human suicide victims has shown reduced NGF mRNA and protein expression in the hippocampus [118]. As such, harnessing and manipulating dysregulated NGF in MDD patients has been of much therapeutic interest. While there is a significant body of work assessing the delivery of NGF via more invasive injectable routes to mouse models of MDD and IN-NGF for other disorders, there has been limited work examining IN-NGF in MDD rodent models.

Preclinical: In one study, mice were administered 2.5 µg IN-NGF (75 µg/kg) and rats 10 µg IN-NGF (50 µg/kg) in staggered doses, 60 min prior to behavioral assessment and post-mortem biochemical analysis. Mice undergoing unpredictable chronic mild stress (UCMS) that were given IN-NGF displayed significantly reduced immobility time in forced swim and tail suspension tests, indicating reduced depressive-like behavior. Decreased sucrose preference (stress-induced anhedonia) in UCMS rats was additionally reversed by 10 µg IN-NGF, which also improved their impaired locomotor activity, and was associated with significantly increased dopamine and norephinephrine levels and enhanced serotonergic (5-HT) receptor expression in the frontal cortex and hippocampus. Immunohistochemistry further showed significantly increased hippocampal neurogenesis and decreased inflammatory markers [119].

### 9.2. IN-NGF Therapies in Neurodegenerative Disorders

#### Frontotemporal Dementia

Clinical: The efficacy of NGF as an IN therapeutic was investigated in two female patients with frontotemporal dementia (FTD) associated with corticobasal syndrome (FTD/CBS). The administration of IN-NGF daily for a year halved cognitive decline with slight improvements in rigidity and word usage that was associated with increased fluoro-deoxy glucose (FDG) uptake by positron emission tomography (PET), indicating increased brain metabolism. Cessation of IN-NGF for one year did not result in any clear return to pre-treatment conditions, but PET images showed significant reduction in FDG uptake and activity [120].

## 10. NPY

Neuropeptide Y (NPY; MW: 4.27 kDa) is a 36 amino acid neuropeptide that is widely expressed throughout the human body. Within the CNS, NPY is predominantly expressed by interneurons of the neocortex, hippocampus, striatum, and amygdala, particularly those also synthesizing GABA and somatostatin. NPY is also present in long projection neurons (e.g., brainstem catecholaminergic groups + arcuate/infundibular hypothalamic nucleus) [121]. NPY binds to Class A rhodopsin-like GPCR receptors located on the cell surface to modulate signaling pathways for cortical excitability, stress response, food intake, circadian rhythms, and cardiovascular function. NPY is released during high-frequency neuronal activity, making it a potent modulator of neurotransmission [121]. The abnormal expression of NPY has been linked to a wide range of disorders, including epilepsy, anxiety and depression, and metabolic disorders. NPY exerts potent neuroprotective effects through the inhibition of cell death, increased trophic support, the inhibition of glutamatergic excitotoxicity, and the normalization of autophagy [122].

### 10.1. IN-NPY Therapies in Neuropsychiatric Disorders

#### 10.1.1. Major Depressive Disorder

Clinical: An initial clinical study assessed the antidepressant efficacy of IN-NPY in stable MDD patients (also on conventional antidepressants), based upon preclinical studies assessing IN-NPY in post-traumatic stress disorder (PTSD) models. IN-NPY (6.8 mg) or placebo was administered adjunctive to this in a double-blind, randomized fashion, with effects assessed at baseline as well as at 1, 5, 24, and 48 h post-treatment. Significant decreases in depression severity in IN-NPY patients were observed at 5 and 24 h post-treatment, with a progressive decrease from the 5 to 24 h mark, though no significant differences from placebo were seen at 48 h [123]. This initial single-dose study shows that the effects of IN-NPY can last up to 24 h, providing a potential framework for dosing frequency in future clinical trials.

#### 10.1.2. Post-Traumatic Stress Disorder

Preclinical: Extensive preclinical studies with IN-NPY have been carried out by Sabban and colleagues, primarily utilizing a single prolonged stress (SPS) paradigm of PTSD in adult Sprague-Dawley rats. Pre-treatment with IN-NPY prior to SPS was found to lessen the severity of stress as assessed by a forced swim test [124]. In a subset of rats sacrificed 30 min post-SPS, IN-NPY significantly reduced elevations of adrenocorticotrophic hormone (ACTH) and corticosterone, and attenuated the induction of tyrosine hydroxylase (TH; a marker of dopamine biosynthesis) in the locus coeruleus. These pre-treatment effects were seen up to 7 days post-SPS, with reduced depressive-like behavior in the forced swim test and anxiety-like behavior in the elevated plus maze [125]. In a subsequent study, rats given IN-NPY for 1 week post-SPS had significantly lower anxiety-like behaviors, as measured by elevated plus maze and grooming, and IN-NPY but not placebo normalized SPS-associated increases in plasma ACTH and corticosterone, and hippocampal glucocorticoid receptor expression [126]. These showed that NPY administration, before or after trauma exposure, ameliorated PTSD-like symptoms and normalized stress-triggered HPA axis dysregulation and noradrenergic hyperactivation.

Subsequent studies have investigated the efficacy of IN-NPY at different timepoints post-SPS, modeling recent trauma exposure versus more developed PTSD-like behaviors and pathophysiology. Treatment with 150 µg IN-NPY immediately following SPS or within 7 days of exposure prevented SPS-induced elevated GR and CRHR protein expression in hippocampus, elevated locus coeruleus norepinephrine levels [127,128]. In behavioral assessments, this dose of IN-NPY was sufficient to reverse symptoms of anxiety (elevated plus maze), behavioral despair (forced swim test), and hyperarousal (acoustic startle response) [128]. With a further delay of 2 weeks post-SPS before initiating treatment with IN-NPY, the proportion of mice displaying severe anxiety (as assessed by elevated plus maze) also increased from 17.5% to 57.1% (indicative of delayed onset/worsening progression of the PTSD-like induced phenotype) and only a higher dose of 300 µg was able to effectively reverse elevated anxiety, depressive-like, and hyperarousal behaviors [128]. However, further testing another 7 days post-NPY revealed that the therapeutic benefit of IN-NPY lasted only 7 days, but that a second IN-NPY dose sustained NPY’s neuroprotective role [129]. Intranasal administration of NPY receptor subtype 1 (Y1R) and 2 (Y2R) agonists to stressed rats revealed that these resiliency effects are dependent on NPY receptor subtype 1 [130]. Additionally, with SPS exposure, there was reduced expression of Y2R in the locus coeruleus that was correlated with increased long-term noradrenergic activation, increased CRH levels in the amygdala, and increased levels of sensitivity of the rodents to mild stressors [125,130,131]. Safety studies assessing the off-target effects of IN-NPY by tracking body weight, food consumption, and cardiovascular responses found that IN-NPY mitigated immediate spikes in heart rate post-SPS but did not otherwise affect cardiovascular functioning or reverse SPS-induced reduced food consumption or weight loss [132,133]. Finally, as all previous studies had exclusively used male rats, female Sprague-Dawley SPS-exposed rats were given placebo, IN-NPY at various doses, or IN-NPY combined with a NPY protease inhibitor immediately following SPS. After two weeks, when behaviorally assessed, only the 600 µg NPY (double effective dose in males) or combined 600 µg NPY with protease inhibitor prevented SPS impairments and depressive-like behavior, indicative of sexually-divergent effects and the role of NPY in the CRH/NPY pathway and HPA axis regulation [134].

Clinical: A pilot clinical study exploring the safety and efficacy of IN-NPY for the treatment of PTSD has been carried out with 24 individuals in a randomized, double-blind, single ascending dose-range study with the following doses: 1.4 mg, 2.8 mg, 4.6 mg, 6.8 mg, and 9.6 mg IN-NPY. Each participant was given single doses of IN-NPY or placebo at least one week apart, with assessments conducted at baseline and following dosing using a trauma script symptom provocation procedure and anxiety testing immediately following trauma script to assess IN-NPY’s effects. No intolerance or significant off-target effects were observed with any dose of IN-NPY, and significant interactions were seen between treatment and dose—with higher doses, more significant anxiolytic effects of IN-NPY were observed in adults with PTSD [135].

### 10.2. IN-NPY Therapies in Neurodegenerative Disorders

#### Huntington’s Disease

Preclinical: NPY has already been demonstrated preclinically and clinically as a potent neuromodulator in PTSD, MDD, and stress disorders, and NPY-expressing striatal interneurons have been correlated with striatal pathology in HD patients and animal models of HD. Therefore, IN-NPY has been explored in preclinical studies using an R6/2 mouse model of HD to further elucidate its role in HD pathology, with 4 week old mice receiving either IN-NPY, IN-NPY Y2 receptor agonist (NPY_13–36_), or saline 5 days a week for a period of 8 weeks. IN-NPY significantly improved motor performance on rotarod testing at the end for treatment, but IN-NPY_13–36_ and saline did not. Both NPY and NPY_13–36_ decreased mutant Htt aggregation, increased dopamine, cAMP regulated phosphoprotein and BDNF levels, and attenuated microglial activation and IL-1β expression [136]. These results indicate that while the Y2 receptor of NPY is involved in both the role of NPY in developing HD pathology and in the therapeutic efficacy of NPY, its activation does not restore HD-deficits, indicating a role for other receptors, such as Y1, which has been shown in preclinical PTSD models to be responsible for the therapeutic benefits of IN-NPY [130].

## 11. Oxytocin

Oxytocin (OXT; MW: 1 kDa) is a peptide hormone with various body-wide functions. Within the CNS, OXT is synthesized within the hypothalamic supraoptic (SON) and paraventricular (PVN) nuclei, and is centrally-released in response to both positive and negative anxiogenic, stressful and social stimuli. Release can be coordinated with vasopressin or occurs independently in a spatial and temporally fine-tuned manner to regulate neuronal processes [137]. OXT is thought to have key roles throughout mammalian evolution in the regulation of complex social cognition and behaviors (attachment, social exploration, recognition, aggression, anxiety, fear conditioning, and fear extinction) [138]. Human behavioral studies have shown that lower levels of plasma and peripheral OXT are found in patients with depression, SZ, and ASD [139,140,141,142]. OXT actions, particularly on the amygdala-cingulate circuit and HPA axis, are thought to be even more significant through their interactions with the IGF-1/growth hormone, dopaminergic, and serotonergic systems.

### 11.1. IN-Oxytocin Therapies for Neurodevelopmental Disorders

#### 11.1.1. Autism Spectrum Disorders

Preclinical: Several studies have investigated the effects of IN-OXT in both WT mice and in various ASD models. One study evaluating the effects of acute versus chronic IN-OXT administration in WT C57Bl/6 mice found that divergent social effects are produced with length of administration. In this study, acute OXT administration resulted in partial increases in opposite-sex social behaviors, whereas chronic administration selectively reduced social behaviors concomitant with a reduction of CNS OXT receptors [143]. General health (e.g., body weight, gross physical appearance) did not change throughout the course of chronic administration, suggesting that prolonged exposure to OXT did not adversely affect the mice.

As clinical research has shown that fetal exposure to valproic acid (VPA) during pregnancy increases the incidence of autism, another group explored the effects of IN-OXT in a prenatal valproic acid-induced ASD mouse model. These mice develop ASD-like behavioral abnormalities and social impairments. A single IN-OXT dose reversed social interaction deficits in these mice for up to two hours. Longer 2-week administration rescued VPA-induced social interaction deficits, as measured against prenatally saline-exposed mice in the social interaction sniff test. The single IN-OXT administration also selectively increased c-Fos expression in the paraventricular nuclei, prefrontal cortex, and somatosensory cortex [144]. Most recently, a group administered IN-OXT to a *POGZ^WT/Q1038R^* ASD mouse model, *POGZ* (pogo transposable element derived with zinc finger domain) being one of the most frequently mutated genes in ASD patients. In mice, this mutation resulted in ASD-like social behavioral deficits, which were restored with OXT administration as measured by the social interaction sniff test. A biochemical analysis also showed that OXT administration restored *POGZ*-reduced levels of oxytocin receptor expression [145].

Clinical: Initial double-blind, placebo-controlled trials administered 12 or 24 IU of IN-OXT or placebo to male children aged 7–16 during parent-child interaction training sessions. Both parent and child behaviors were assessed beforehand and at multiple timepoints during the live-in intervention—primarily social interaction skills, repetitive behaviors, and emotion recognition. No significant improvements with IN-OXT were found [146].

Nevertheless, a subsequent study was performed by another group looking at changes in brain activity during judgements of socially and non-socially meaningful pictures in 17 children with ASD before and after IN-OXT administration. Measurements were taken using functional magnetic resonance imaging (fMRI) and found that IN-OXT significantly increased activity in the striatum, middle frontal gyrus, medial prefrontal cortex, right orbitofrontal cortex, left superior temporal sulcus, and left premotor cortex. IN-OXT enhanced activity for social stimuli and attenuated activity during nonsocial judgments [147]. A subsequent IN-OXT neuroimaging clinical trial by the same group examined 21 children with ASD (age 8–16.5, 18 of 21 male). Participants blindly received either IN-OXT or placebo prior to an initial fMRI scan. On a subsequent visit, the children received the other agent before a second scan. IN-OXT enhanced activity in brain regions important for social-emotional connection, specifically the superior temporal sulcus (pSTS), amgydala, nucleus accumbens, and prefrontal regions [148]. The effects of IN-OXT on reward circuity responses in ASD children was similarly examined: 28 children with ASD completed two fMRI scans, either after IN-OXT or placebo. During both sessions, participants completed social and nonsocial incentive delay tasks. Increased brain activation following IN-OXT administration was only seen during nonsocial reward anticipation conditions, during which there was greater activation of the right nucleus accumbens, left anterior cingulate cortex, bilateral orbital frontal cortex, left superior frontal cortex, and right frontal pole [149].

The effects of IN-OXT on nonsocial aspects of ASD (e.g., restricted interests and repetitive behaviors) as determined by an automated eye-tracking task were also examined. ASD participants fixate on more highly organized and structured (systemized) images, whereas control subjects showed no gaze preference. IN-OXT eliminated this preference in ASD participants, but in control participants, it increased their visual preference for more systemized images. The authors concluded that the effects of IN-OXT may extend beyond social communication in alleviating some of the nonsocial deficits experienced with ASD [150].

In adults with ASD, IN clinical trials have explored the short-term effects of single doses of OXT and longitudinal studies have examined the long-term therapeutic benefit to social and cognitive functioning. Initial studies paired adult men with ASD with age-matched healthy controls and administered 24 IU IN-OXT or placebo before and after real-time, naturalistic social interaction simulations, finding that IN-OXT significantly enhanced eye gaze and eye contact in both ASD patients and healthy controls, with the most significant effect noted on participants with ASD and/or those with the most impaired levels of eye contact at baseline [151]. Subsequent studies examined the dose-dependent effects of 8 versus 24 IU IN-OXT on social-cognitive deficits in adults with ASD, finding that there were no significant differences between administration of either OXT dose shortly before a series of social-cognitive tasks, though both were significantly more effective than placebo in the modulation of overt emotional salience [152]. Further studies investigated IN-OXT as an adjuvant to social reinforcement behavioral therapy, finding that with 20 IU IN-OXT, ASD individuals showed enhanced learning during social learning targets and when receiving social feedback (as compared to non-social) [153]. With 24 IU IN-OXT, the perception of dynamic and static social vs. non-social stimuli, an ability negatively associated with ASD, increased significantly, particularly with gaze preference towards the eyes of fearful faces [154]. Most recently, however, in a single 24 IU dose IN-OXT fMRI imaging study, no significant changes in brain activity during behavioral tests following OXT administration were found, despite increased bilateral amygdala responsiveness with OXT during the physical pain task [155].

Longitudinal studies looking at the effects of IN-OXT on social-cognitive deficits experienced with adult ASD have ranged in treatment span from 4–12 weeks. In adult men with ASD, once-daily self-administered doses of 24 IU IN-OXT for a period of 4 weeks significantly improved mood states and reduced avoidance and self-reported repetitive behaviors [156,157,158]. Salivary samples collected from the OXT treatment group at baseline, end of treatment period, four weeks after, and one year after, found that endogenous OXT levels remained higher 4 weeks post-treatment, indicating that repeated IN-OXT administration can induce long-lasting changes in endogenous OXT levels, likely through a self-perpetuating feedforward elevation of OXT correlated with the positive effects of OXT on social behaviors [157]. Functional MRI imaging at baseline following a single dose of IN-OXT found it to attenuate bilateral amygdala activity, and multi-dose treatment for 4 weeks to induce a consistent attenuation in brain activity which outlasted the treatment period even at four weeks and one year post-treatment [158]. Consistent with the aforementioned single-dose studies, the greatest attenuations of amygdala activity, increases in salivary OXT levels, and greatest behavioral improvements (particularly with avoidant attachment behavior and social functioning) were experienced by those participants with the greatest impairments at baseline [156,157,158]. A longer 24-week clinical trial in young adults with either ‘high-functioning’ ASD or pervasive developmental disorders (>15 years of age) was conducted with a 12-week double-blind period in which participants received a high dose (32 IU), low dose (16 IU), or placebo followed by a 12-week open-label period in which all participants received the higher dose. A significant improvement in symptoms was observed only in male participants (the study did not have adequate female participants for statistical analysis). Furthermore, in analyzing single nucleotide polymorphisms of the *OXTR* gene, it was found that the efficacy of chronic OXT administration is modulated by dosage (≥21 IU) and a specific SNP of *OXTR* (rs6791619) [159].

#### 11.1.2. Schizophrenia

Clinical: Similar to IN-OXT clinical trials with ASD, both single-dose studies and multi-dose longitudinal clinical trials have been done for SZ, but unlike with ASD, these studies have only been explored in schizophrenic adults because the DSM-V reports adolescent presentation as rare.

With single-dose studies, a multitude of doses have been explored, with clear divergence in results for different doses. Earlier studies investigated the effects of varying, singly-administered IN-OXT doses prior to behavioral tasks. With 10 versus 20 IU in SZ patients with or without symptoms of polydipsia, a symptom affecting 20% of SZ patients, and with increased hippocampal-modulated AVP and stress hormone responses to psychological stress, all patients under 10 IU had reduced emotion. However, with 20 IU, emotion recognition improved, but only in polydipsic SZ patients, thought to primarily be due to the divergent effects on the bias in identifying fear in nonfearful faces [160]. A subsequent study examined recognition of kinship and intimacy in SZ patients versus healthy controls with the administration of 24 IU IN-OXT or placebo 45 min prior to engaging in social interpersonal tasks [161]. IN-OXT improved complex social perception in both healthy subjects and SZ patients, but recognition of kinship only improved significantly with OXT in the SZ patient group [161].

Additional studies investigated effects on social cognition of single dose 40 IU IN-OXT or placebo administered to adult male veterans with SZ, before and after which participants completed lower- and higher-level assessments of social cognition (facial affect perception, social perception, detection of lies, detection of sarcasm and deception, and empathy). Though there were no significant composite score changes between the OXT and placebo groups, higher-level social cognitive task subscores significantly improved for the OXT group compared to placebo, suggesting that IN-OXT effects may primarily be on higher-level social cognitive impairments in SZ [162]. These findings were supported by another study that also administered 40 IU IN-OXT or placebo to male SZ patients and age-matched healthy controls to assess IN-OXT effects on autonomic (ability to interpret/understand emotional cues) versus controlled (ability to comprehend indirectly expressed emotions, thoughts, and intentions through complex deliberations) social cognition. At baseline, SZ patients showed significant impairments in both autonomic and controlled social cognition, and IN-OXT administration significantly improved only controlled but not autonomic cognition deficits, with limited effects on social cognition in healthy controls [163]. Further double-blind studies at the same dose in age-matched male SZ adults and healthy controls investigated facial cognition, changes in bilateral amygdala activity, and eye fixation [164,165]. IN-OXT decreased amygdala activity for images of fearful faces, increased activity for happy faces, attenuated amygdala activity in SZ patients, and increased activity in healthy controls [164]. SZ patients had reduced eye fixation time with placebo that then increased with IN-OXT administration, which was predicted by higher attachment anxiety and greater symptom severity, while in the control group, IN-OXT decreased eye fixation time in comparison to placebo [165].

Single-dose clinical studies done with doses above 40 IU have not reproduced the trend of previous studies in finding benefits of IN-OXT to ameliorate deficits in adult SZ. The administration of 48 IU IN-OXT to SZ patients and age-matched controls followed by facial emotion matching and control tasks failed to show any improvement in facial affect processing, contradicting previous results with IN-OXT [166]. Similarly, the administration of 50 IU IN-OXT to SZ patients and age-matched controls found no improvements of impaired judgement in SZ patients [167].

IN-OXT supplementation to behavioral therapy and cognitive skills training has also been investigated. In one study, IN-OXT was combined with 12 sessions of social cognitive skills training (focused facial affect recognition, social perception, and empathy) over the course of 6 weeks. SZ patients were randomly assigned (double-blind) to receive either 50 IU OXT or placebo 30 min prior to each session. At the end-of-treatment and 1 month post-treatment, subjects getting OXT had significantly greater improvements in empathetic accuracy than those taking placebo, but OXT-related effects on other social cognitive domains were not observed [168]. In a longer clinical trial supplementing IN-OXT to a 24-week (48 session) cognitive behavioral social skills training (CBSST) program, male adults with SZ or schizoaffective disorder were randomized to either 72 IU OXT or placebo. At the end of the 24-week period, contrary to results from the previous 6-week study, no significant differences were found in any of the parameters measured (social functioning, positive symptoms, negative symptoms, defeatist beliefs, or asocial beliefs) [169].

The large number of promising clinical studies with IN-OXT has fueled a lot of excitement about its therapeutic potential, but many IN-OXT studies with similar study designs to those discussed above have found little to no observed effects on social cognitive impairments. These include a 3-week IN-OXT study [170], a 6-week trial comparing IN-OXT to galantamine (only galantamine treatment was associated with improved negative symptoms) [171], a longer 12-week 48 IU daily IN-OXT trial that found limited benefit of OXT on introspection and PANSS reduction of negative subscores, but no other effects on outcome measures [172,173], and an 8-month treatment with 40 IU IN-OXT daily, that found no significant effects on clinical symptoms or psychosocial behavior [174].

### 11.2. IN-Oxytocin Therapies for Neuropsychiatric Disorders

#### 11.2.1. Post-Traumatic Stress Disorder

Clinical: Clinical trials with IN-OXT have been explored in two contexts: (1) recently trauma-exposed individuals with the intent of modulating the consolidation of trauma associated memories to prevent development of PTSD, and (2) individuals diagnosed with clinically-significant PTSD with the intent of attenuating developed symptoms and exacerbated anxiety. The earliest clinical trials administering IN-OXT to recently trauma-exposed people utilized a randomized double-blind placebo-controlled setup with a single dose of 40 IU IN-OXT prior to fMRI imaging. Baseline functional connectivity was assessed in 37 participants 11 days post-trauma with a focus on emotion regulation areas [ventromedial prefrontal cortex (vmPFC), ventrolateral prefrontal cortex (vlPFC)], and salience processing areas [insula, dorsal anterior cingulate cortex (dACC)]. IN-OXT reduced amygdala-left vlPFC activity after trauma script-driven imagery, compared with neutral script-driven imagery and against placebo. Additionally, IN-OXT generally increased amygdala-insula activity while decreasing activity in the amygdala-vmPFC region [175,176]. These neural effects resulted in lower levels of sleepiness after the trauma script in participants given IN-OXT but not placebo, suggesting that OXT may interfere with emotion regulation functioning following trauma reminders in recently-trauma exposed individuals. In an emotional faces processing task combined with fMRI imaging before and after IN-OXT or placebo, OXT significantly increased right amygdala activity, and there were also significant sexually divergent effects of OXT in responses to neutral faces, to which women also showed increased left amygdala activity with IN-OXT [176]. Contrary to these results, a subsequent randomized double-blind placebo-controlled clinical trial, wherein IN-OXT was administered daily within 12 days of trauma for 8 days and assessments, conducted at baseline and 1.5 months post-trauma, found that IN-OXT reduced PTSD, depression, and anxiety, even at follow-ups 6 months post-trauma [177,178]. However, these effects were only seen in patients with higher symptom severity scores at baseline, while patients with lower baseline scores showed no clinically significant improvements [177,178].

In similar neuroimaging studies of police officers diagnosed with PTSD compared to healthy controls, IN-OXT’s effects on amygdala reactivity (measured by fMRI) towards facial affect recognition, in social incentive delay, in monetary incentive delay, and in distraction-emotional regulation tasks, were assessed as measures of PTSD symptoms and anhedonia [179,180,181,182]. In PTSD patients, the increased amygdala activity towards fearful-angry faces as compared to happy-neutral faces was absent, and 40 IU IN-OXT dampened amygdala activity towards all emotional faces in both male and female participants. In PTSD patients with greater anxiety and amygdala reactivity prior to IN treatment, IN-OXT administration had more significant reductions in amygdala activity [179]. PTSD patients under placebo showed reduced left anterior insula responses to social rewards that were increased with IN-OXT, which also increased responses to social reward in the right putamen [180]. In the monetary incentive delay task, which assesses the anticipation of monetary reward and loss as a measure of anhedonia, IN-OXT increased neural responses in the striatum, dorsal anterior cingulate cortex and insula during reward and loss anticipation in both PTSD patients and controls. Though no significant differences were seen in PTSD patients and controls in reward processing, severity of PTSD anhedonia was negatively correlated to ventral striatum reward responsiveness, which IN-OXT did increase [181]. In an emotion regulation strategy task, IN-OXT enhanced left thalamus activity during distraction in both PTSD and healthy patients, but specifically enhanced left thalamus-amygdala coupling during distraction in PTSD patients, indicating that IN-OXT may improve cognitive emotional regulation abilities in PTSD patients [182].

In an exclusively-female PTSD cohort paired with age-matched healthy controls, 24 IU IN-OXT or placebo was administered prior to fMRI imaging to determine efficacy in reducing the intensity of provoked PTSD symptoms, which was assessed by Facial Affect Recognition and cardiac control. IN OXT reduced total PTSD symptoms, particularly avoidance, from baseline fMRI scanning, which was accompanied by a decrease in maximum heart rate and a drop in the pre-ejection period, a marker for improved sympathetic cardiac control. There was a positive correlation between endogenous OXT levels and decreased heart rate, showing for the first time that IN-OXT can reduce the intensity of symptoms in female PTSD patients [183]. Possible benefits of IN-OXT in a combined evidence-based prolonged exposure therapy setting has also been tested in PTSD patients: 17 individuals with diagnosed PTSD of diverse origin self-administered 40 IU IN-OXT or placebo 45 min prior to weekly exposure therapy. PTSD patients receiving OXT had trends of lower PTSD and depression symptoms during exposure therapy and higher working alliance scores, though the study failed to achieve statistical significance, suggesting possible benefits for IN-OXT as augmentation for PTSD therapies [184]. In a separate neuroimaging study with a subset of male and female PTSD patients who had experienced childhood abuse, 24 IU IN-OXT was assessed for its ability to modulate PTSD-related aberrant threat and reward processing. In the facial affect recognition task, IN-OXT reduced left amygdala-left/right anterior insula connectivity for women in response to fearful faces, but increased left amygdala-right anterior insula connectivity in men while also reducing right amygdala-right anterior insula connectivity. These results suggested that IN-OXT’s mechanism in modulating threat salience in childhood trauma-exposed individuals can vary as a function of sex [185].

#### 11.2.2. Generalized/Social Anxiety Disorders

Clinical: The earliest clinical trial for social anxiety disorder explored IN-OXT as an adjunct to exposure therapy. A total of 25 male participants with diagnosed social anxiety disorder (SAD) self-administered 24 IU IN-OXT or placebo prior to each of 5 weekly group sessions. Participants who received IN-OXT showed improved positive evaluations of appearance and speech performance, but no significantly altered overall treatment outcome, suggesting the limited effects of IN-OXT on mental representations of self and social anxiety [186].

Subsequent studies investigated the effects of IN-OXT on social behaviors and cognition during social rejection, as well as on reward motivation. To simulate OXT’s ability to modulate perceptions of social rejection, 60 men with primary diagnoses of SAD were administered either 24 IU IN-OXT or placebo prior to social exclusion tests. Only individuals with low attachment avoidance displayed more social affiliation and cooperation under OXT, whereas those with high attachment avoidance had faster detection and reaction to disgusted and neutral faces [187]. When a separate cohort of 52 males with SAD received 24 IU IN-OXT or placebo prior to a reward motivation task to measure willingness to work for self versus others’ monetary needs, a similar trend was seen in that less socially anxious participants who received OXT indicated greater willingness to work for self, and highly socially anxious participants did not have any beneficial effect from IN-OXT. This led the group to conclude that the ability of OXT to modulate levels of social anxiety may only go so far, and to work for less severe presentations of SAD [188].

The effects of IN-OXT in generalized anxiety disorder with social anxiety (GSAD), which is characterized not only by aberrant patterns of amygdala-frontal connectivity to social signals of threat, but also at rest, were examined in an fMRI imaging study. Modulation of fear reactivity following administration of 24 IU IN-OXT or placebo to GSAD patients and matched healthy controls prior to an emotional faces paradigm was assessed. GSAD patients had heightened medial PFC and anterior cingulate cortex activity in response to fearful and sad faces, which was significantly reduced to the level of controls by IN-OXT. These results indicated a specific modulatory effect of IN-OXT on fear-related amygdala reactivity and non-threatening negative social cues [189,190]. A subsequent trial with similar design, administering 24 IU IN-OXT or placebo in GSAD patients and healthy matched controls [191,192], found that IN-OXT enhanced functional connectivity between the amygdala, bilateral insula, and middle cingulate/dorsal anterior cingulate gyrus in response to fearful faces, in individuals with GSAD only [191]. Additionally, OXT enhanced the functional connectivity of the left/right amygdala with the rostral anterior cingulate cortex in combination with medial PFC in response to fearful faces, essentially reversing GSAD-induced deficits in amygdala-frontal connectivity to the levels of the healthy controls [192].

### 11.3. IN-Oxytocin Therapies for Neurodegenerative Disorders

#### Frontotemporal Dementia

Clinical: As there are social, cognitive, and behavioral deficits associated with multiple variants of FTD, particularly bvFTD, IN-OXT, which has shown much therapeutic promise in treating social and behavioral deficits in SZ, ASD, and PTSD, has also been explored for treatment of social behavioral deficits in FTD. In initial trials to determine safety and tolerability, patients with bvFTD or semantic dementia were given doses of either 24, 48, or 72 IU IN-OXT or placebo, twice daily for 1 week. All three doses were well tolerated by patients, and testing suggested beneficial effects of IN-OXT in treating behavioral symptoms, specifically apathy and empathy [193,194]. In a subsequent single-dose imaging study, 28 FTD patients received 72 IU IN-OXT or placebo prior to completing an fMRI combined facial expression mimicry test. Neural activity in brain regions associated with empathy, emotion processing, and the stimulation network were recorded. IN-OXT administered alone or IN-OXT combined with the mimicry task increased activity in regions of the simulation network and in limbic regions associated with emotional expression processing, demonstrating the efficacy of IN-OXT in ameliorating affected limbic and other frontotemporal regional deficits during social cognition in FTD patients [195].

## 12. PACAP

Pituitary adenylate cyclase-activating polypeptide (PACAP; ADCYAP; MW: 4.5 kDa) is a neuropeptide hormone of the VIP/secretin/glucagon/growth hormone-releasing hormone family [196]. Within the CNS, PACAP is expressed primarily by nuclei of the brainstem, hypothalamus and thalamus, amygdala, cerebral cortex, medulla oblongata, and posterior pituitary as a 178-amino acid preprohormone. In the peripheral/autonomic nervous system (ANS) [197], PACAP is the primary neurotransmitter expressed by preganglionic neurons at the sympathetic adrenomedullary synapse. The precursor prohormone is processed by prohormone convertases (PC1 and PC2), primarily producing PACAP38 (38-amino acid residues in length) and tissue-specific PACAP27 (MW: 3.1 kDa). PACAP functions as a neurotransmitter, neuromodulator, hypophysiotropic hormone, and neuroprotective agent, and is activated in the regulation of stress responses, cardiovascular function, food intake, circadian rhythm, and reproduction [198]. PACAP binds to three G-protein coupled receptors: PAC_1_ (most widely expressed in CNS), and VPAC1/2 (wider peripheral tissue expression). PAC_1_ receptor binding can activate a wide variety of downstream signaling pathways, including adenylyl cyclase, phospholipase C, MEK/ERK, and AKT, regulating hormonal secretion, energy metabolism, and neuronal survival. PACAP has also been shown to regulate glial glutamate turnover by regulating the expression of glial glutamate transporters GLT-1 and GLAST, as well as glutamate-converting enzyme (GS) through PAC_1_ receptors.

### 12.1. IN-PACAP Therapies for Neurodegenerative Disorders

#### 12.1.1. Alzheimer’s Disease

Preclinical: Progressive decline of endogenous PACAP levels correlates with the pathological severity of cognitive decline in post-mortem analysis of human MCI and AD patient brains [199]. Many studies have therefore explored the local and systemic administration of PACAP in rodent AD models, with varying degrees of success [200]. A single study has investigated IN-PACAP in an APP transgenic mouse model (APP_V7171_), with the goal to harness PACAP’s potent α-secretase activating efficacy for nonamyloidogenic processing of APP to, in turn, indirectly reduce Aβ expression and oligomeric deposition. Three-month old male mice were given 10 µg IN-PACAP daily for 3 months, which stimulated nonamyloidogenic processing to lower Aβ expression, and lowered BDNF and Bcl-2. In Novel Object Recognition testing, mice given PACAP had significantly less cognitive impairment, with recognition indices close to that of wildtype controls [201]. Overall, IN-PACAP administration significantly ameliorated Aβ-associated deficits in this mouse model, but as manipulation of APP secretases for AD therapies has been shown to be insufficient in reversing decades-long plaque deposition with clinically-significant patient benefits, it remains to be seen how effective IN-PACAP will be.

#### 12.1.2. Huntington’s Disease

Preclinical: The role of PACAP and the PAC_1_ receptor in HD is not well understood, but with post-mortem human HD hippocampal samples showing specific decreases of PAC_1_ without changes in VPAC_1_ or VPAC_2_, its involvement and possible therapeutic efficacy have been explored in recent years by Xifró and colleagues in R6/1 mouse models of HD. In a preclinical study, 13-week-old R6/1 and wild-type mice were given 30 µg/kg IN-PACAP (38 AA isoform) daily for a week, followed by behavioral and biochemical analysis. In novel object recognition and T-maze testing, R6/1 mice given IN-PACAP had significantly reduced memory deficits and increased hippocampal VGlut-1 and PSD95 expression [202]. At the onset of motor symptoms, R6/1 mice displayed reduced striatal expression of PAC_1_R levels, which IN-PACAP restored, while it additionally restored levels of CREB-binding protein and BDNF [203].

## 13. Discussion

We have reviewed a growing body of preclinical and clinical studies conducted over the past two decades that have examined the development and efficacy of IN CNS therapeutics. These peptides provide a promising and fast-developing option for safe, non-invasive, targeted therapeutics. They are small in size, have the ability to utilize olfactory/trigeminal innervations and cribiform gaps to cross the BBB, and their hydrophilic nature allows them to retain bioactivity in the aqueous CNS environment. In terms of off-target effects, only a limited number of these IN trials found any of note, attesting to the targeted nature of this route of delivery. Additionally, even with “short” treatment periods, such as with OXT and NPY, subjects remarkably continued to show efficacy even a year post-administration. Many of these IN therapeutics also offer an advantage over many oral small-molecule therapeutics in their ability to target protein-protein interactions (PPIs), long regarded as an “undruggable space”. Overall, this method of delivery has led to revived interest in long-abandoned therapeutics, such as GDNF, which had previously failed in 1990s clinical trials that utilized invasive intracranial injections. Altogether, this review presents an overview of the rapid progress that has been made turning a technique, proposed just a few decades ago, into a viable clinical method for safe, targeted delivery of CNS therapeutics.

The therapeutic outcomes of each intranasal therapy are complex, dependent on dose and treatment length, and highly variable for a single peptide administered for different disorders. Although many of the preclinical and clinical IN studies reviewed here have presented encouraging data in human subjects and animal models, the vast majority of these studies do not contain statistically significant numbers of females, making general conclusions about efficacy difficult. Justification for the selective inclusion of male subjects has included (1) the complexity introduced by the female menstrual cycle, (2) simplicity of all-male study designs, and (3) the ease of comparison of results to previous male studies. However, other papers cited herein [184,185] have demonstrated that by scheduling all women in their study for IN-OXT dosing during the luteal phase of their cycle, they were able to successfully control for menstrual cycle variations in OXT response [116,117]. The US National Institutes of Health (NIH) has long-recognized the importance of, and since 2016 has mandated, the accounting of sex as a biological variable in NIH-funded human and animal studies [204]. We therefore hope that future studies and trials exploring the development of IN CNS therapeutics will be more inclusive of sex in study design.

We have summarized the clinical and preclinical progress of each IN therapeutic for a given disorder in Table 1 (next page).

## Figures and Tables

**Figure 1 cells-11-03629-f001:**
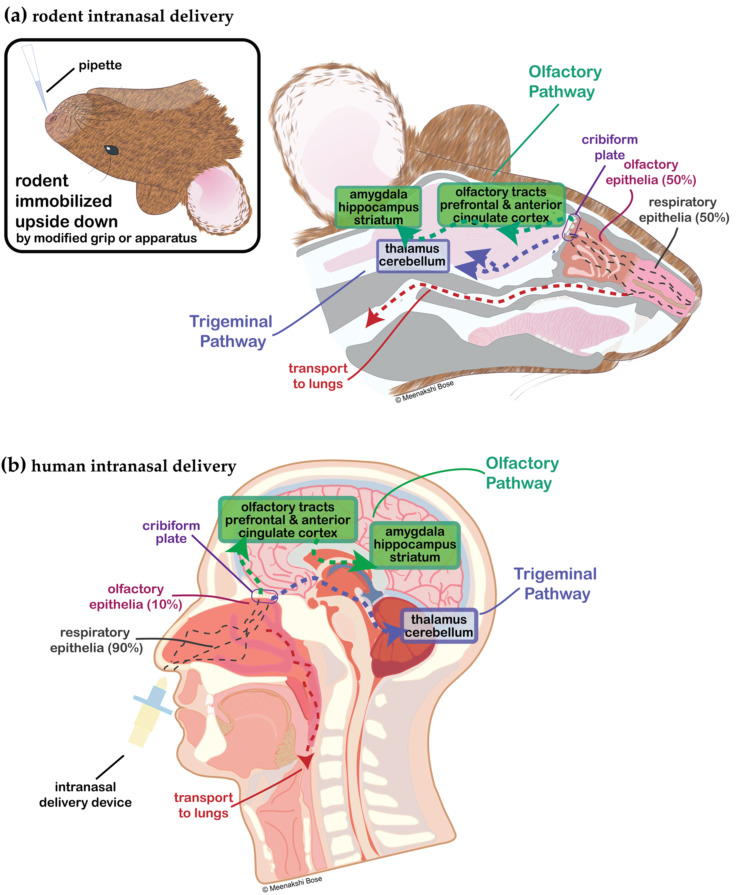
Diagrammatic representation of intranasal administration in (**a**) rodent and (**b**) human. (**a**) Rodent is held upside down in a modified one-handed grip and low a volume-high concentration peptide drug is delivered to awake (non-anesthetized) animals [24]. Rodent nasal epithelium is 50% respiratory epithelia and 50% olfactory epithelia, whereas (**b**) human is 90% respiratory and 10% olfactory. With human administration of intranasal drugs targeted to the CNS, a variety of nasal delivery devices are utilized in clinical trials to target delivery to the upper nasal cavity and olfactory epithelia, maximizing nose-to-brain transport across the cribiform plate [25].

**Table 1 cells-11-03629-t001:** **Summary table of all significant intranasal peptide drug data** as reviewed from literature, preclinically in rodent models and/or clinically in randomized controlled trials of varying design. Effects seen listed as: **⇑** = improved; **⇓** = reduced; **⊕** = restored; **∅** = eliminated/prevented. Subsections: NDD = neurodevelopmental disorders; NPD = neuropsychiatric disorders; ND = neurodegenerative disorders. Disorders: ASD = Autism Spectrum Disorder; BD = Bipolar Disorder; SZ = Schizophrenia; MDD = Major Depressive Disorder; GSAD = Generalized/Social Anxiety Disorder; PTSD = Post-traumatic Stress Disorder; AD = Alzheimer’s Disease; PD = Parkinson’s Disease; HD = Huntington’s Disease; FTD = Frontotemporal Dementia.

Peptide	Disorder		Preclinical (Rodents)	Sample Size	References	Clinical	Sample Size	References
**BDNF**	NPD	MDD	**⇑**hippocampal BDNF **⇑**body weight **⇓**anxiety-like behaviors	80–83	[26,27]	--	--	--
**EPO**	ND	AD	**⇑**locomotor activity, cognition, memory **⇓**microglial activation **⇓**TNF, Bax/Bcl2-ratio, Fas, IL-1β **∅** hippocampal neurodegeneration	84–154	[30,31]	--	--	--
**GDNF**	ND	PD	**⇑**locomotor activity **⇓**microglial hyperinflammation **⇓**lesion severity (40%) **∅**6-OHDA induced weight loss **⊕**dopaminergic neurons	16–53	[36,37,38]	--	--	--
**GLP-1**	ND	AD	**⇑**spatial learning and memory **⇓**Aβ plaque formation **∅**circadian rhythm disturbances	3–8	[40,41]	--	--	--
**Insulin/IGF-1**	NDD	ASD	** -- **	--	--	**⇑**social and cognitive improvement **⇓** restrictive behaviors in children <3yo	9–25	[49,50]
		BD	--	--	--	No significant effects	62	[52]
		SZ	--	--	--	**⇑**POMC expression	39–45	[53,54,55]
	NPD	MDD	--	--	--	No significant effects	35	[56]
		GSAD	--	--	--	**⇓**skin conductance response **⇓**saliva and plasma cortisol	26–123	[61]
	ND	AD	**⇑** glucose uptake/CNS insulin signaling **⇑**cognition **⇑** neurogenesis **⇓** Aβ, tau **⇓**microglial activation	7–44	[65,66,67,68,69,70]	**⇑** cognition **⇑** working memory/recall **⇓**peripheral insulin resistance	12–289	[71,72,73,74,75,76,77,78,79]
		PD	**⇑**motor performance **⇓**ODHA neurotoxicity/DAergic neuron loss **⊕** CNS insulin/Akt/GSK3β signaling	8–39	[80,81,82]	**⇑** motor performance **⇑** FAS word score **⇓** parkinsonism severity	8	[83]
		HD	**⇑**locomotor activity **⇑**cortical & striatal Akt/mTOR signaling **⇓**mHtt phosphorylation	32	[84]	--	--	--
**NAP**	NDD	ASD	**⇑**cognition **⇓** inflammatory cytokines (TNF, IL-6, IL-12) **⊕**structural abnormalities **∅**irregular immune activation	12–28	[87,92,93,94]	--	--	--
		SZ	**⇑**cognitive function **⇓** anxiety **⊕**Foxp2 expression	61	[95,96,99]	**⇑**functionally-significant cognition	63	[100]
	ND	AD	**⇑** cognitive performance **⇓**Aβ_40_/Aβ_42_, tau **⇓**anxiety-like behaviors **∅**loss of choline acetyltransferase activity	20–35	[101,102,103,104,105,106]	No significant effects	144	[107,108,109]
		PD	**⇑**locomotor activity **∅**dopaminergic neuron loss	56	[110,111]	--	--	--
**NBD**	ND	AD	**⇑**spatial learning and memory **⇓**Aβ plaque formation **⇓**neurodegeneration **∅**NF-kB activation, neuroinflammation **⊕**CREB, mGluR1 expression	24	[117]	--	--	--
**NGF**	NPD	MDD	**⇑**hippocampal neurogenesis **⇓**inflammatory markers **∅**stress-induced anhedonia **⊕**cortical & hippocampal 5-HT expression	24	[119]	--	--	--
	ND	FTD	--	--	--	**⇑**word usage **⇓**rigidity	2	[120]
**NPY**	NPD	MDD	--	--	--	**⇓**MDD severity (on MARDS) at +5 and +24 h	30	[123]
		PTSD	**⇓**depressive and anxiety behaviors **∅**stress-induced increases in ACTH, corticosterone, hippocampal glucocorticoids **∅**hyperarousal	18–36	[122,125,126,127,128,129,130,131,132,133,134]	**⇓**anxiety (BAI)	24	[135]
	ND	HD	**⇑**locomotor activity **⇓**mHtt phosphorylation **⇓**microglial hyperinflammation	10	[136]	--	--	--
**Oxytocin**	NDD	ASD	**⇑**c-Fos in PVN, PFC, and SSC **⇑**opposite-sex social behaviors **⇓**(selective) OXTr expression **⊕***POGZ*-reduced OXTr expression	21–44	[143,144,145]	**⇑**striatal, prefrontal, and motor activity **⇑**social-emotional cognition **∅**eye fixation anomalies **⊕**bilateral amygdala activity	17–38	[146,147,148,149,150,151,152,153,154,156,157,158,159]
		SZ	--	--	--	**⇑**controlled cognition **⇑**empathetic accuracy **⇑**outcomes with combined skills therapy **⊕**bilateral amygdala activity	23–68	[160,161,162,163,164,165,166,167,168,169,170,171,173,174]
	NPD	PTSD	--	--	--	**⊕**multi-regional amygdala connectivity	34–107	[175,176,177,178,179,180,181,182,183,184,185]
		GSAD	--	--	--	**∅**abnormal PFC and ACC activity **⊕**amygdala-frontal connectivity	36	[186,187,188,189,190,191,192]
	ND	FTD	--	--	--	**⇑**frontotemporal and limbic connectivity **⇑**empathy **⇓**apathy	25–60	[193,194,195]
**PACAP**	ND	AD	**⇑**nonamyloidogenic processing **⇓**BDNF, Bax/Bcl2-ratio **⊕**cognitive function	14–30	[201]	--	--	--
		HD	**⇑**locomotor activity **⇑**spatial learning and memory **⇑**hippocampal VGlut1,PSD95 **⊕**BDNF, CREB-binding protein, PAC1r	24	[202,203]	--	--	--

## Data Availability

Not applicable.

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
