# Peer review of "Intranasal Peptide Therapeutics: A Promising Avenue for Overcoming the Challenges of Traditional CNS Drug Development"

_cells, 2022, doi:10.3390/cells11223629_

Round 1

Reviewer 1 Report

The review covers the mass of studies both preclinical and clinical for treatment of central nervous system diseases using small molecules delivered intranasally. While the introduction focuses on the lack of research for peptides as therapeutics in CNS diseases, it would benefit from a reduction in language and run-on sentences. The review is currently split by disorder but may be easier to follow if outlined as in the abstract by the twelve peptides reviewed. This review comes from a  well-known lab with expertise in the field and does an excellent job of highlighting the preclinical and clinical studies thus far. One caveat to most of the experiments are the small study sizes which might be highlighted in an additional table unless in table 1 (table 1 could not be reviewed, was not attached to document).

1.    Minor:
Revise for run-on sentences throughout, fix nomenclature (if capitalized in one section maintain throughout eg Schizophrenia/schizophrenia). Line 126 suggest change “mutated genes” to “variants. Add citations to line 63, 129, 321

Major:
Overall the review is a behemoth of data and suggests IN administration of small molecules is an underserved avenue of therapeutics, however, much of the studies reviewed showed little or specific groups benefit. Adding a section on the metabolism of small molecules or different mechanisms of IN delivery (are they all in saline? Etc) would benefit the review. Line 99 Revise- schizophrenia and bipolar are not largely known to be due to genetic origin but a complex of genetic, social, and environmental triggers whereas NDDs are present from birth (entirely genetic in cause). Table 1 missing. 

Reviewer 2 Report

In the present manuscript, the authors review the current preclinical and clinical literature to support the use of intranasal peptide therapeutics for an array of neurodegenerative, neuropsychiatric, and neurodevelopmental disorders.  The work is dense, but well written, and catalogs the efficacy of several promising peptides in a host of CNS disorders.  A particularly useful summary table is provided at the conclusion of the work.  There are some issues with the work, both major and minor, that need to be addressed.  Perhaps the biggest issues are that the work would benefit greatly from editing to reduce the amount of material describing each CNS disorder and the methodological details of the studies described, and the inclusion of more information regarding potential problems associated with such therapeutics, both with regard to their immunogenicity/off-target effects and their CNS penetrance.

MAJOR POINTS

1. The work would benefit greatly from editing to condense the amount of material describing each CNS disorder and reduce the excessive description of the methodologies employed in these studies.

2. While it is stated in the Discussion section that peptide treatment strategies are “safe, non-invasive, targeted therapeutics” and that in the studies described “..off-target effects, only a limited number…found any”, more information regarding potential problems associated with such therapeutics should be included with regard to their immunogenicity, off-target effects, and their CNS penetrance.

3. Figure 1 not particularly useful in its present form.  The labeling font is difficult to read in Panel B, labeling is not provided in Panel A, and a key should be provided to denote respiratory and olfactory epithelia.

4. Important information is missing (e.g. “name of model”) on lines 907, 914, 948, 979, and 1033.

MINOR POINTS

1. By convention, “TNF-alpha” is now named “TNF” as “TNF-beta” is now known as “lymphotoxin-alpha”.

2. Line 585, caspase-3 is referred to as an “inflammatory marker” but is typically considered to be an apoptotic (and hence immunoquiescent) marker.

3. The document includes inappropriate use of capitalized words that are not proper nouns.

4. There are syntax issue on Lines 48-50, 1102-1103, and 1166

5. There are missing grammatical articles (e.g. Lines, 210, 857, 1102, 1105, and 1106).

6. There is missing punctuation Lines, 347, 929, 1100, 1135, 1146, and 1147.

7. It is not clear what is meant by “has yet to be published” on Lines 957 and 1010.  Is this intended to mean that further studies are required?

8. It is unclear what is meant by “survival of the SN dopamine” Line 1062 and “challenging beam and pole testing” Line 1083.

9. All abbreviations should be defined the first time that they are employed.

Round 2

Reviewer 1 Report

line 276 add citation

line 482-483 remove new paragraph

line 762-763 remove new paragraph

line 839 recommend add " schizophrenic adults because the DSM V reports adolescent presentation as rare." 

line 876 change "contrasting" to "contradicting" 

line 965/966 change "gender" to "sex"

line 1084 citations are not correct format

line 1090 change "gender" to "sex"

line 1093 add citations

line 1095-1097 simplify sentence

line 1098-1099 move to end of paragraph one in discussion line1076

Table 1: add citations column vs section column for ease of readers

Author Response

  1. line 276 add citation

response: The following citation has been added.

Calkin, C.V., Insulin resistance takes center stage: a new paradigm in the progression of bipolar disorder. Annals of Medicine, 2019. 51(5-6): p. 281-293.

  1. line 482-483 remove new paragraph

response: This correction has been made.

  1. line 762-763 remove new paragraph

response: This correction has been made.

  1. line 839 recommend add " schizophrenic adults because the DSM V reports adolescent presentation as rare." 

response: This addition has been made.

  1. line 876 change "contrasting" to "contradicting" 

response: This correction has been made.

  1. line 965/966 change "gender" to "sex"

response: This correction has been made.

  1. line 1084 citations are not correct format

response: Citations have been corrected for formatting.

  1. line 1090 change "gender" to "sex"

response: This correction has been made.

  1. line 1093 add citations

response: The following citation has been added.

Arnegard, M.E., et al., Sex as a Biological Variable: A 5-Year Progress Report and Call to Action. Journal of Women's Health, 2020. 29(6): p. 858-864.

  1. line 1095-1097 simplify sentence

response: This sentence has been simplified.

  1. line 1098-1099 move to end of paragraph one in discussion line1076

response: This paragraph has been moved as requested.

  1. Table 1: add citations column vs section column for ease of readers

response: The section column has been removed and two reference columns (one preclinical, one clinical) have been added with comprehensive citing of all primary research articles covered in the review.

Reviewer 2 Report

The authors have significantly improved the manuscript.  The work remains exhaustive, but has benefited from editing and a reorganized presentation of the material.  The discussion of the immunogenicity/off-target effects and the CNS penetrance of these agents remains scant, but this appears to be largely due to the lack of consideration of such factors in the literature.  The figure provided is markedly improved and the other errors noted have been corrected.

Author Response

Response: We thank the reviewer for their positive comments on the revised manuscript.